# Simulation of Corrosion Phenomena in Automotive Components: A Case Study

**DOI:** 10.3390/ma16155368

**Published:** 2023-07-31

**Authors:** Annalisa Ferrarotti, Elisa Vittoria Ghiggini, Riccardo Rocca, Matteo Dotoli, Federico Scaglione, Claudio Errigo, Giancarlo Marchiaro, Marcello Baricco

**Affiliations:** 1Department of Chemistry, NIS-INSTM, University of Turin, Via Pietro Giuria 7, 10125 Torino, Italy; annalisa.ferrarotti@unito.it (A.F.); elisa.ghiggini@gmail.com (E.V.G.); riccardo.rocca@unito.it (R.R.); matteo.dotoli@unito.it (M.D.); federico.scaglione@unito.it (F.S.); 2Centro Ricerche Fiat, C.R.F. S.C.p.A, Corso Settembrini 40, 10135 Torino, Italy; claudio.errigo@crf.it (C.E.); giancarlo.marchiaro@crf.it (G.M.)

**Keywords:** corrosion, computer simulations, coating materials

## Abstract

Mathematical modelling and software simulation nowadays are very effective tools for both understanding and predicting corrosion processes and the protection of metallic components. COMSOL Multiphysics 5.6 software provides validated mathematical models that can be used, for a given geometry, as a tool to predict and prevent corrosion of components. The corrosion of zinc-coated steel sheets has been studied in this work by comparing results of the simulations with laboratory tests carried out in a salt spray. Results of both the mathematical modelling and empirical tests give the possibility to estimate the stability of the protective zinc layer over time. The examination of the discrepancies between two analytical methods for the investigation of corrosion phenomena leads to possible modifications in the model in order to reach as much as possible coherence with experimental data. As a final result, a computational model of corrosion phenomena in an automotive component has been reached, allowing in the future to partially substitute laboratory tests, usually being highly time consuming and expensive.

## 1. Introduction

The study of corrosion phenomena is of primary importance in the manufacturing industry, both concerning structural components and functional ones. Corrosion in coated metallic components occurs as a deterioration of products caused by the interaction with external agents (soil, atmosphere, water). Corrosion of metallic materials can be general or localised, depending on which part of the product is subjected to deterioration, and its understanding and prevention is fundamental in order to avoid possible extensive damages [1]. In the automotive sector, attention is focused mainly on prevention of possible corrosion phenomena affecting both parts of the vehicle exposed to atmospheric agents and those possibly subjected to other types of aggression, including relatively high thermal or mechanical stresses [2,3]. Studies of these phenomena are usually carried out through empirical tests. Among them, the accelerated corrosion tests, simulating “real-life” mechanisms, find a wide range of employments. Specifically, in the automotive industry, accelerated corrosion tests involve mostly analyses of the components undergoing salt spraying and humidity exposure [4,5]. In salt spray tests, performed in specific experimental chambers, the salt solution is nebulised onto the automotive component, simulating outdoor conditions of exposure to weathering, and degradation of the substrate is measured as a function of time. During the exposure to salt fog, the testing atmosphere is static, maintaining a single experimental environment, with selected values of both temperature and humidity [4,5,6]. Humidity interaction is often also studied, reflecting service conditions of a vehicle, by the application in specific experimental chambers of controlled humidity levels, typically varying from 60% to 95%. Very often, humidity cycles are alternated with drying cycles and combined with a temperature variation in order to predict alterations of the system in highly critical conditions. The aim of these tests is to measure the thickness of the forming oxide layer or the thinning of the metallic coating at different time intervals. Apparently, these methods are simple and relatively inexpensive, however, they are extremely time consuming, as some components need to be studied for long periods [4,5,7]. For the reasons mentioned above, the simulation of corrosion phenomena based on thermodynamic and electrochemical rules and intrinsic material properties has become increasingly important today and is often used as an alternative to or in support of experimental testing. Simulation of corrosive phenomena is useful because, firstly, it can be used to make preliminary assessments of the most suitable materials for the required application and, secondly, because it can replace or support empirical tests to assess the corrosion resistance of the components [8,9].

One of the main approaches to simulate corrosion phenomena in manufactured components is the FEM (Finite Elements Method) [8,10]. The analyses based on this method are built starting from the assumption that a complex component can be divided into many small domains (the so-called “finite elements”), within which simple equations, related to thermodynamic, kinetic or transport phenomena, can be applied to describe its chemical and physical behaviour. The global properties of the complex object are therefore given by the sum of all the individual finite elements, behaving in a certain way, into which the object is subdivided [10]. Several software, based on the FEM to solve the constituent partial differential equations, are distributed nowadays, and one of the most popular is COMSOL Multiphysics [11]. The term “multiphysics” indicates that many physical mechanisms can be combined and implemented to describe accurately complex real phenomena. The workflow of this software starts with the choice of the input system on which the analysis must be applied; a pre-existing file, built with other modelling or technical drawing programs (AutoCAD 2023), can be imported, or the component can be drawn directly in the software by using suitable implemented tools. Then, the second step is the selection of both materials and physical phenomena to be modelled, with the assignment of appropriate parameters to each component of the system. After these steps, the construction of the mesh is necessary to reduce the external substrate into finite elements, and the size of the mesh can be established by the user or even automatically defined by the software. The finer the mesh, the more accurate and punctual the simulation will be; however, this is at the expense of time and computing resources necessary for the calculation. Then, the software applies the study to the components and the calculator will return graphs and tables that can be interpreted and analysed directly on the COMSOL Multiphysics 5.6 software or by other spreadsheet-based programs [8,9,10,12].

An FEM approach was employed by Forrest et al. [13] for the determination of a method for cathodic protection of bronze propellers on a Cu-Ni surfaced slip. In this study, some experiments have been conducted in a water tunnel in order to establish the need for cathodic protection by the insertion of individual anode locations and multi-anode systems. Another use of the FEM as a support of measurements has been performed by Kasper et al. [14,15], who measured the polarisation curves of the constituent materials and then compared the results with the model that calculated near field potential and current density distributions. Moreover, the localised corrosion has been simulated by Sharland et al. [16], in the case of the presence of isolated cavities on the metal surface, using the FEM as a support to solve the complex set of mass-conservation equations describing the system, and aiming to develop a mathematical representation of the physical mechanisms controlling the process. In the early 2000s, Cui et al. [17] computationally studied the ability of wetted 316 L stainless steel cathodes to support a stable localised corrosion site in the case of exposure to thin electrolytes. They studied the effect of parameters, both physical and electrochemical, on the total net cathodic current, aiming to predict the stability of localised corrosion sites. Mandel et al. [18] studied the galvanic corrosion behaviour of an aluminium/CFRP (carbon fibre-reinforced plastic) laminate self-piercing rivet joint and an aluminium/steel blind rivet joint. They performed polarisation measurements in a 5%wt NaCl solution, and they used these data as boundary conditions for the finite element simulations. Then, they concluded that the highest corrosion rates were found at the material interface, linked to the highest calculated potential gradients. Later, zinc-coated steel sheets were analysed by Saeedikhani et al. [19] in order to understand the mechanism of corrosion protection in the case of damage of the coating, simulated by a scratch. The results showed a sacrificial protection in the middle of the scratch, which has been predicted by simulations in terms of zinc consumption. More recently, finite element simulations, using COMSOL Multiphysics 5.6 software, have been performed as support and comparison with experimental data for the study of galvanic corrosion between Al–Zn–Mg–Cu alloy and stainless steel in the salt-spray atmosphere by Peng et al. [20] in order to individuate the extent of pitting corrosion in different regions of the component.

The aim of this paper is to apply the FEM, using the COMSOL Multiphysics software, to predict the extent of the corrosion over time in a zinc-coated sheet of steel, as a reference of components employed in the automotive sector. The defined zinc-coated sheet of steel simulates the exterior of cars, daily exposed to weathering and consequently subject to corrosion. The output of major interest is the change in thickness of the zinc coating, experimentally evaluated by salt spraying the metallic sheets for different time frames. The experimental results have been subsequently compared to the simulated outcomes, i.e., COMSOL Multiphysics calculations, to evaluate analogies and discrepancies in the model and possibly to modify the simulation in order to find more reliable input parameters. Therefore, apart from the above-mentioned thickness change of the coating, which is the most relevant output given by the software, a study of influencing parameters has been conducted in order to understand more deeply possible correlations between the input parameters and the final results.

## 2. Materials and Methods

Corrosion effects have been studied on a sheet of steel (BH210) of rectangular shape (100 mm × 200 mm) with a zinc coating with a thickness of 7 µm. Five holes with 10 mm diameter have been created in the sheet for a better understanding of the corrosion behaviour near boundaries, along which it is likely to be concentrated. Firstly, experimental tests, i.e., potentiodynamic polarisation measurements, have been conducted onto the substrate (i.e., a steel sheet without coating) and on the coating (i.e., a zinc sheet) in order to extract necessary electrochemical input parameters for the simulation study. Potentiodynamic polarisation measurements involved measurements of the total corrosion current density as a function of the applied potential and the corrosion potential when the material was in contact with an electrolyte, at room temperature. The investigated material was included in a conventional three-electrode configuration set up system, where it acted as “working electrode”, together with a Pt inert counter electrode and an SCE (Saturated Calomel Electrode) “reference electrode”. Starting from the open circuit voltage (OCP), stabilised in 12 h, the working electrode was scanned from −1.1 V to +1.0 V at a scanning rate of 0.5 mV/s in 5%wt NaCl solution in water. The ionic conductivity of the electrolyte was equal to 7 S/m, as reported in the technical sheet. Then, accelerated corrosion tests were performed, nebulising the same NaCl solution onto the sheets in order to see the effect of the salt fog aggression and to simulate actual weathering. The experimental chamber used for testing the components is SC/KWT 1000, produced by Weiss Technik, and the tests were performed according to ISO 9227 and ASTM B117 normative.

The experimental conditions include a fixed temperature of 35 °C, a relative humidity of 95% and a pH of 6.5. Tests have been followed by both visual inspections, to estimate the evolution of corrosion processes, and the measurements of coating thickness variation over time. The latter has been evaluated using X-ray fluorescence (XRF), which is a non-destructive technique useful for measuring, with high precision, both the coating thickness and chemical composition of single and multi-layered coatings. The instrument used for the measurements is the X-STRATA 980 of Oxford Instrument and the reference standards used for testing are in according with DIN ISO 3497 and ASTM B 568 normative.

The computational investigation was performed using COMSOL Multiphysics 5.6 software. The system was modelized as a rectangular sheet with five holes, made of a substrate of steel and a coating of zinc. The same dimensions of the samples used for experimental salt fog measurements have been considered. The selected model system was assigned to a CAD description to be used for the mesh definition. Appropriate materials in the COMSOL database were selected and, for each of them, experimental polarisation curves have been loaded. In building the simulation model, the input parameters have been introduced, matching the experimental conditions, while electrochemical properties of involved elements have been extracted from the literature, see Table 1.

The time evolution of the system was studied, measuring the variation of the coating thickness at regular time intervals of 24 h, up to 168 h, in 24 points diffused on the metallic sheets in different regions of interest, as shown in Figure 1. The three main regions on which the attention of the study is focused are the inner area of the sheets (i.e., point n. 15, 16 and 17), the region along the external boundaries (i.e., point n. 14, 18, 19, 20, 21, 22, 23, 24) and the regions near the holes (i.e., point n. 1, 2, 3, 4, 5, 6, 7, 8, 9, 10, 11, 12, 13).

The selected straightforward output (i.e., the thickness variation of the zinc coating) is provided by COMSOL in the form of both visual representation of the thinning in a coloured scale and a table of values for each point at every time step; the first output allows a quick overview of the most corroded regions, while the second gives more quantitative details on the entity of corrosion process in the selected points of interest.

## 3. Results and Discussion

Firstly, from the OCP measurements, the open circuit voltage has been extracted after a stabilisation of 12 h, and it stabilised around −1.005 V. Then, two polarisation curves were measured before starting the simulations for both materials involved, and results are reported in Figure 2, with steel in Figure 2a and zinc in Figure 2b. The polarisation curves, which substantially describe the electrochemical behaviour of the material at different voltage or current conditions, can be considered as the main sensitive parameters for the method. Indeed, these curves are crucial input parameters inserted in the simulation, so they must be measured empirically before starting calculations with COMSOL Multiphysics 5.6 software.

From the analysis of the polarisation curves, some interesting information can be extracted, e.g., corrosion potential and corrosion current density. From the polarisation curve of zinc, which is more interesting, being the coating of the exposed surface to corrosion, these values are, respectively, −1.01 V and 2.2 · 10^−4^ A/cm^2^. The polarisation curves for both materials have been shown as measured, represented by the black curve, and after the increase of the current density of 10% (in blue), 20% (in green) and 30% (in red). As discussed in the following, the variation of the current density has been calculated in order to investigate possible discrepancies between experimental and calculated data, considering the different experimental conditions used for potentiodynamic polarisation measurements and for salt fog tests.

The result of the first simulation (i.e., considering experimental data for the polarisation curves, black curves in Figure 2 of the thickness variation as a function of the exposure time to the salt fog) is shown in Figure 3. It can be noticed that, as expected, the corrosion process is progressive over time and the thickness of the surface coating degrades more rapidly along external boundaries and near the holes. Corrosion, in general, becomes consistent after 72 h, when the red-coloured zone (i.e., corresponding to a coating layer fully corroded), starts to involve not only peripheric regions. In fact, corrosion starts preferentially where the electrochemical reactions are favoured to take place, e.g., defects, corners, asperities, in this case represented by the edges and the holes of the sheets.

A comparison of both experimental and simulated results is shown in Figure 4a–c, which reports experimental data on points (see Figure 1) near the boundaries, near the holes and inner points, respectively; (d), (e) and (f) report data simulated with COMSOL Multiphysics on the same positions. For experimental data referring to points near edges and near holes (Figure 4a,b), the general trend is rather similar, showing a corrosion process starting with a visible decay in thickness change up to 70 h, then the thinning becomes less prominent, reaching a plateau value near 8 µm, corresponding to the total consumption of the coating thickness (7 µm) and a starting corrosion of the core material. In Figure 4c, referring to internal points, the overall trend is similar to those previously described, but with larger fluctuations in the time interval between 50 and 70 h. Comparing the simulated results with the experimental data, some differences are clearly visible. In Figure 4d,e, the thinning proceeds slowly up to 50 h, then, in the time range between 60 and 120 h, a sudden decrease of the coating is visible, reaching a final plateau corresponding to its total consumption. In Figure 4f, the thickness change is close to zero up to 120 h, then a very slight variation can be observed.

Discrepancies between experimental and calculated data can be primarily attributed to the fact that, in simulating the process, COMSOL assumes ideal surfaces without any defects. This is a crucial aspect, because corrosion mechanisms are usually triggered by imperfections, asperities related to roughness and, more generally, to high energy surfaces, of which simulated sheets are lacking [1]. Another observation, useful to understand the differences between the two sets of results, is that in experimental data, the thickness variation is related to the initial thickness of the zinc coating, which was not the very same for each sample sheet. Therefore, an error is associated with the results, as represented in Figure 4a–c by error bars.

For a visual comparison between simulated and experimental results, corrosion evidence is reported for selected time intervals in Figure 5, starting from the initial situation at 0 h, with a fully cyan-coloured sheet and a fully zinc-coated sheet of steel. After 48 h of exposure to aqueous sodium chloride solution, external edges of the coated sheet start corroding, as it is observed in the simulated sheet by the red-coloured region. Comparing the sheet exposed to the salt fog after 48 h, it is visible that there is a large corrosion along the external edge at the bottom of the sheet and a more uniform corrosion on the other regions. Then, after 120 h, both external edges and regions near the holes appear to be corroded in simulated data, while, even in this case, the experimentally tested sheet shows a more uniform corroded condition. Finally, after 168 h, the result of simulation shows a homogeneous corrosion of the whole surface, except for the internal region, which is in contrast with the experimentally tested sheet, showing a distribution of corroded surfaces. In general, the simulation for high time frames demonstrates an advancement of the corrosion and, consequently, a decrease in the thickness of the protective zinc layer. However, the boundary conditions neglect the triggering of corrosive phenomena in specific points of the lamination.

Together with possible roughness and asperities spread onto the surface of samples [1], another hint that might explain observed discrepancies is that, in corrosion simulations, potentiodynamic polarisation curves performed with an electrolyte solution at room temperature are considered, whereas different experimental conditions (i.e., temperature and electrolyte) have been used for salt fog tests. For this reason, current density was considered to vary by 10%, 20% and 30%, as previously shown in Figure 2. An overview of the coating thickness reduction, after considering a variation in the current density, is shown in Figure 6. The main aspect to be highlighted is that, as expected, triggering of the corrosive processes occurs earlier in time, and it is of increased entity. In fact, in the simulated sheets at 168 h, it is clearly visible that the larger the current, the smaller the non-corroded region, cyan-coloured.

In order to highlight variations in corrosion times led by variations in the polarisation current density, the thinning of the coating layer for three significant points is reported in Figure 7. As expected, after increasing the current density, the decay in thickness for each point is shifted to lower time steps with respect to the original curve. In general, the larger the current density variation, the earlier the decay of the thinning curve. Apart from this general trend, it is straightforward to notice that the effect of current density variation on each point is not the same. In fact, in Figure 7a, corresponding to point 9, placed near a hole, by adding a 10% of current density, basically no variations can be detected, while an acceleration of corrosion becomes consistent from 20% current increase. On the contrary, in Figure 7b, corresponding to point 14, placed near the lateral edge of the sheet, the effect of current variation displayed is nearly the same for each current addition. This non-homogeneous effect on the surface is related to a non-homogeneous distribution of the process onto the sample. Even if current density is significantly increased up to 30%, corrosion effects remain quite limited in internal points, as observed in Figure 7c, corresponding to point 15. Overall, by increasing the current density, the model is approaching experimental data, represented in Figure 7 by the pink lines.

Finally, some information about the halving time can be extracted for different significant points spread through the surface of the zinc-coated steel sheet. Concerning the region near boundaries, i.e., point 14, the result obtained considering the experimental current densities leads to 21.3 h of halving time, which is strongly different from the simulated one, which turned out to be about 80.1 h. With an incremental percentage of polarisation current density, see Figure 2, the halving time decreases progressively. In fact, for a 10% increment, it corresponds to 78.7 h, for 20% it is of 62.1 h and for 30% it is of 56.8 h. Considering point 9, near the holes, a similar trend is observed. In fact, the experimental halving time is 22.4 h, while the originally simulated one is around 70.9 h. After adding an increment of polarisation current density values, it became equal to 57.3 h for a 10% increase, 57.0 h for 20% and 56.3 h for 30%. The trend is similar, being the experimental halving time similar for both points, while those calculated after the current density increment show a similar decrease in the halving time. As can be seen in Figure 8, while for the point placed near the boundaries (point 14), a significant decrease of halving time is observed, increasing the current density, for the point near the hole (point 9), when increasing the polarisation current density, the variation in the halving time is quite limited. Linearly fitting the two simulated curves, it has been possible to extrapolate the percentage of current density increment to match the experimental values of halving time. These are 100% and 70% for point 9 and 14, respectively. It is worth noting that for starting experimental values of the polarisation curve, in order to join calculated and experimental results obtained in a different environment, a different increment of current is necessary for different points of the component.

Lastly, considering the internal point (point 15), the experimental halving time is 21.9 h, similar to values of the other considered points. However, the calculated values are strongly different, showing values approximately near zero at the end of the tests, also for simulation involving a current density with a 30% increment. This result can be explained considering the approximation of the zinc-coated steel sheet as an ideal surface, so discrepancies between calculated and experimental results become likely more evident in internal points of the sheet because this approximation is more unrealistic in this region.

For both points 9 and 14, a further variation in current density has been performed according to the trend simulated in Figure 8 in order to make the simulated results fit better with experimental results. In the simulations performed using these extrapolated values of current density, all the input parameters used in the simulation have been kept unaltered, as reported in Table 1 of the manuscript. Therefore, the only values that have been varied are those of current density, referring to the polarisation curve being considered, as explained in the manuscript, which are the most sensitive input parameters for the simulation. The results of the estimation of the thickness variation after the increment of the current density (i.e., 100% and 70% for point 9 and 14, respectively) are reported in Figure 7a,b with the light blue curve. As expected, the curves better fit the experimental results in terms of halving time, but the behaviour remains similar to those of simulated curves, so it slightly differs from that of experimental tests. This result suggests that in order to better reproduce the corrosion rate, the change of further parameters should be considered.

## 4. Conclusions

The corrosion processes in zinc-coated steel sheets have been studied, comparing a computational approach by means of finite element method simulations using the COMSOL Multiphysics software, with an experimental approach, with accelerated corrosion tests. The main parameter used to evaluate the lifetime of the components, subjected to accelerated corrosion conditions (salt fog 5%wt NaCl), simulating actual weathering conditions, was the variation of the thickness of the zinc coating protective film from the initial value of 7 µm. The thinning process, evaluated as a function of time, returned data predicting a gradual corrosion, with both preferential initiation and higher speed in proximity of edges and near the holes, but with lower speeds in the internal region of the component. From experimental tests, the halving time of the zinc coating is approximately the same for all the considered points of the sheet, being around 20 h. Conversely, from the simulation results, the thinning is calculated at higher times, and it strongly depends on the position of the examined point. Furthermore, different simulations have been performed with a current density increment of 10%, 20% and 30%, observing that the larger the current density increment, the lower the halving time. Comparing results of simulations with those of experimental tests in different points of the sheet, it is worth to notice that the resulting trend in the thinning is slightly different. In addition, in order to match simulated and experimental results, a different variation of the current density should be considered. These observations can be explained by the fact that simulation does not consider the effects of surface roughness on corrosion, assuming the surfaces as ideally flat. However, it is known that the actual roughness of the material surface can trigger electrochemical reactions, so the corrosion mechanism can start not only from the edges of the considered object, but also from the internal points.

## Figures and Tables

**Figure 1 materials-16-05368-f001:**
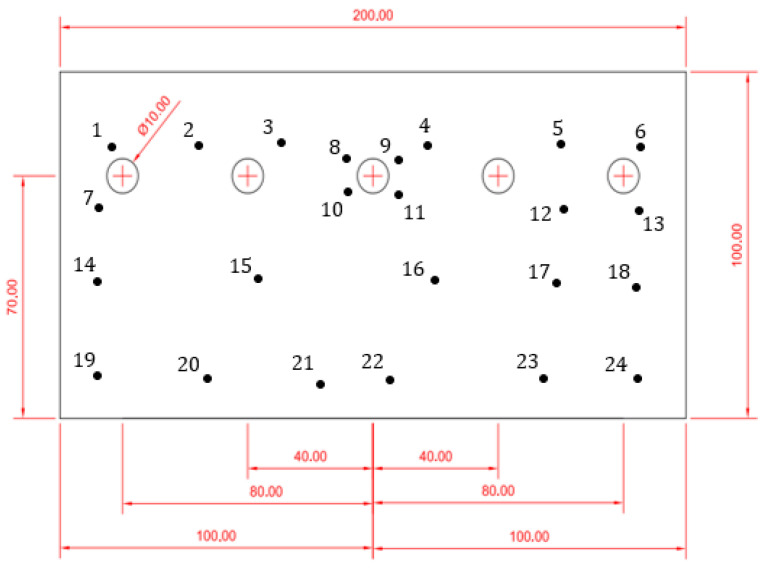
Representation of the selected points of interest on the zinc-coated sheets of steel. Reported values are in mm.

**Figure 2 materials-16-05368-f002:**
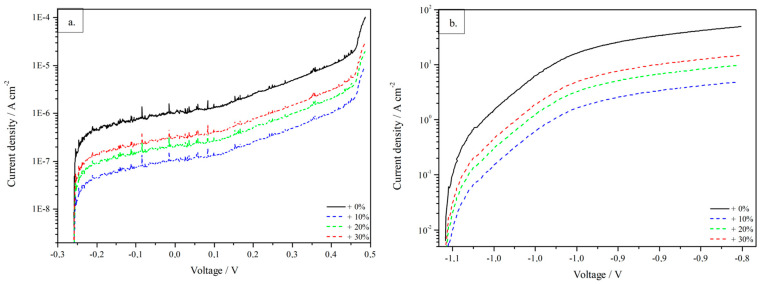
Experimental polarisation curves (black) of the materials constituting the sheet, (**a**) curve referring to the core material (steel BH210) and (**b**) curve referring to the coating material (zinc). Possible variations of current at fixed voltage are represented by coloured curves (blue + 10%, green + 20%, red + 30%).

**Figure 3 materials-16-05368-f003:**
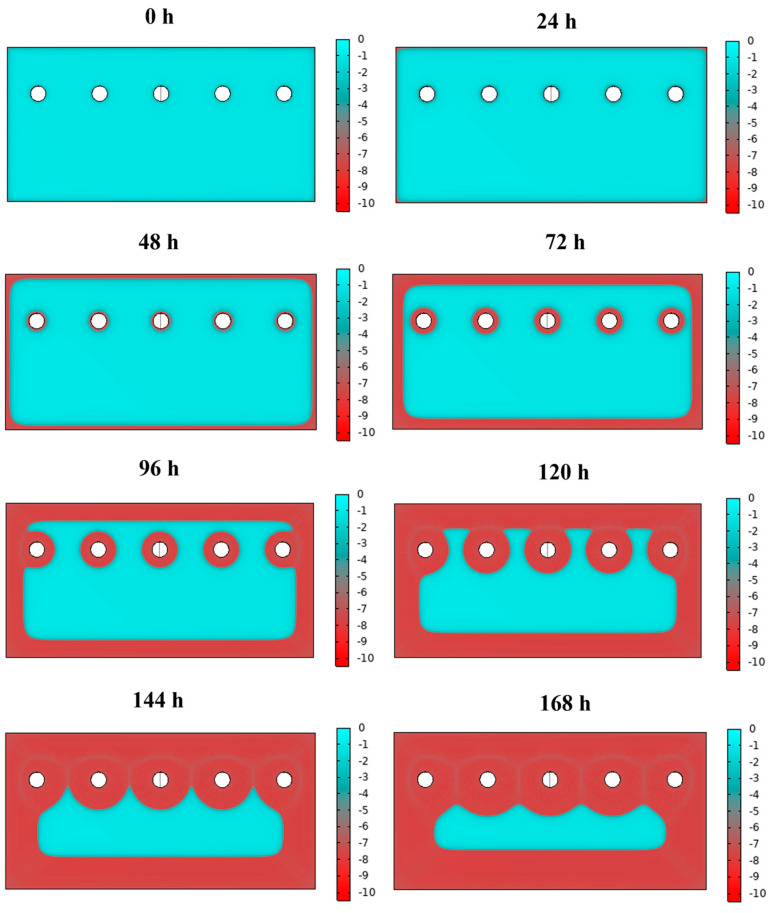
Results of the corrosion simulation on a zinc-coated steel sheet from the initial situation equal to 0 h up to 168 h. The thickness in micrometres of the Zn surface layer is shown in coloured scale (cyan: non-corroded; red: fully corroded).

**Figure 4 materials-16-05368-f004:**
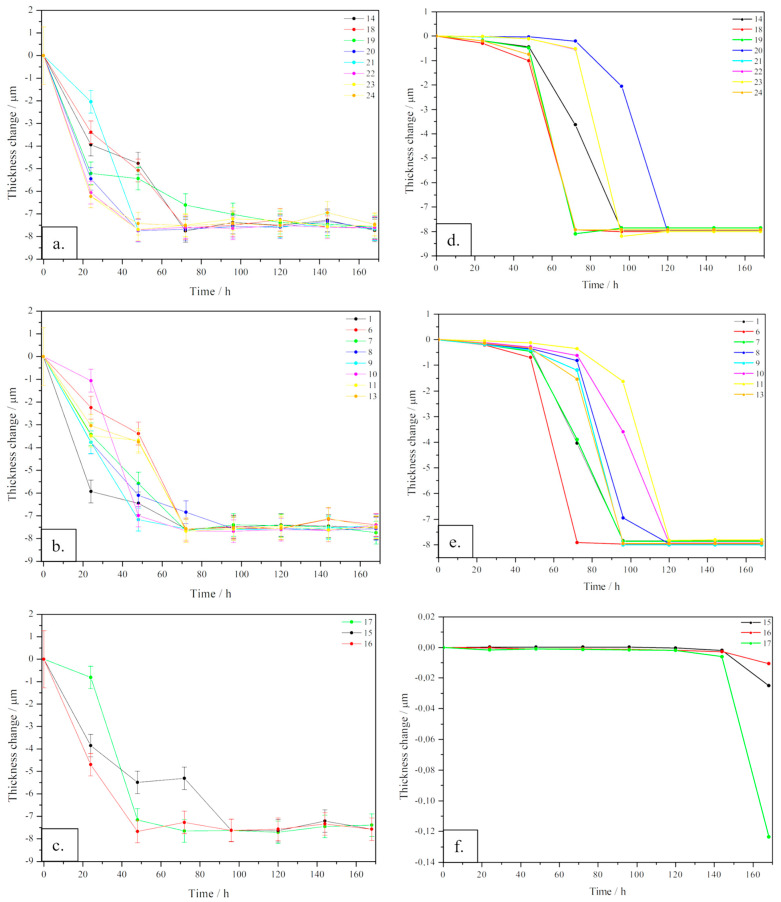
Comparison of results of the corrosion simulation on zinc-coated steel sheets with experimental results, deriving from salt fog experimental tests, in different regions of the sheets. (**a**–**c**) report experimental data on points near the boundaries, near the holes and inner points, respectively; (**d**–**f**) report data simulated with COMSOL Multiphysics, on points near the boundaries, near the holes and inner points, respectively.

**Figure 5 materials-16-05368-f005:**
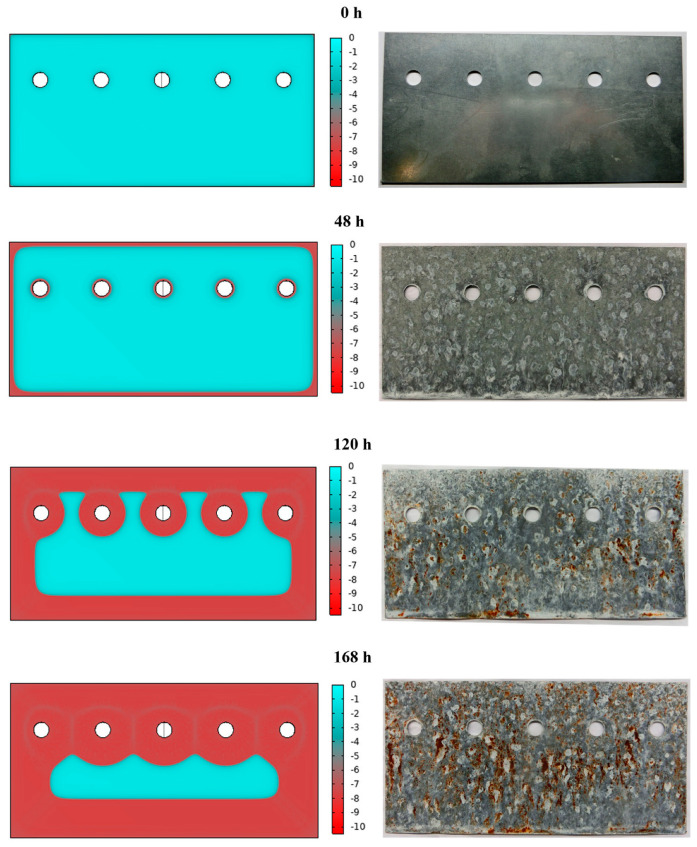
Comparison of (**left** side) results of the corrosion simulation on zinc-coated steel sheets with (**right** side) experimental results, deriving from salt fog experimental tests, at different time intervals. The thickness variation of the Zn surface layer is shown in coloured micrometres in scale (cyan: non-corroded; red: fully corroded).

**Figure 6 materials-16-05368-f006:**
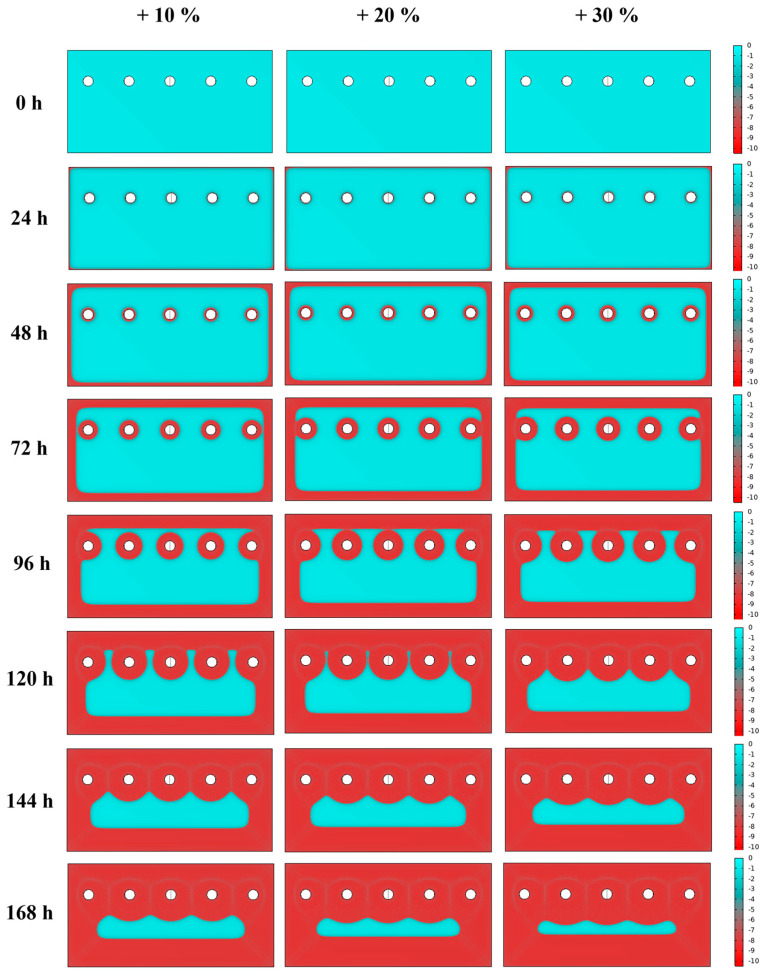
Comparison of results of the corrosion simulations on zinc-coated steel sheets from the initial situation equal to 0 h up to 168 h, increasing the experimental value of the current density to 10, 20 and 30%. The thickness of the Zn surface layer is shown in coloured micrometres scale (cyan: non-corroded; red: fully corroded).

**Figure 7 materials-16-05368-f007:**
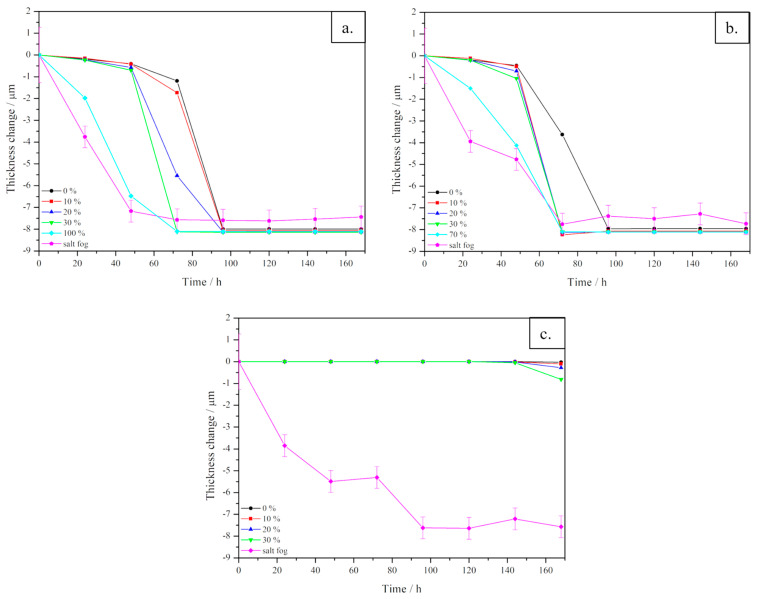
Comparison of the results of the corrosion simulation with the variation of current density and experimental curve, in three different points of the zinc-coated steel sheets, located in different regions. (**a**) corresponds to point 9 placed near a hole, (**b**) corresponds to point 14 near the lateral edge of the sheet and (**c**) corresponds to point 15, an internal point.

**Figure 8 materials-16-05368-f008:**
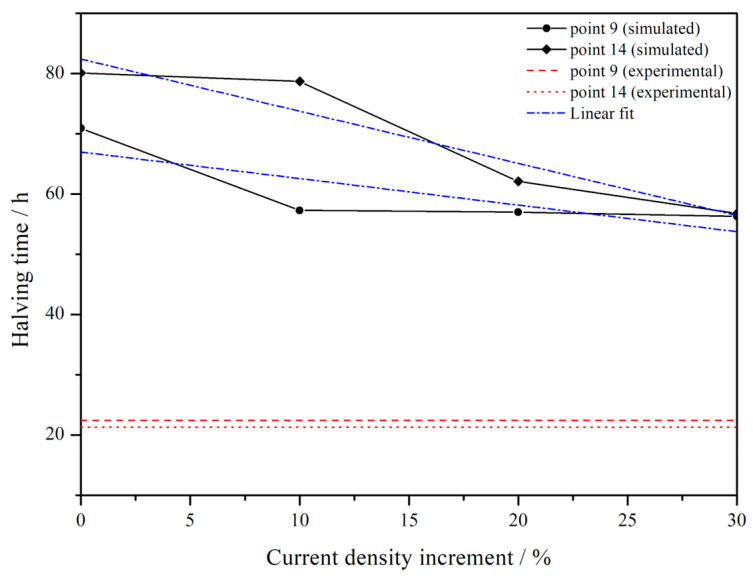
Halving time as a function of current density increment of point 9 and 14. The linear fit of experimental data is shown as dashed-dot lines. For comparison, the experimental values of halving time are shown as red lines.

**Table 1 materials-16-05368-t001:** COMSOL input parameters. The values marked with * are the arbitrary established initial conditions of the simulation, the others have been taken from the literature.

Parameter	Value	Description
Sigma *	7 S/m	Electrolyte conductivity
s_change *	−7 × 10^−6^ m	Zn film thickness
rho_Zn [21]	7130 kg/m^3^	Zn density
RH *	0.95	Relative humidity
M_Zn [21]	0.06538 kg/mol	Zn molecular weight
Eeq_Zn [21]	−0.76 V	Equilibrium potential of Zn
Eeq_O_2_ [21]	1.23 V	Equilibrium potential of O_2_
Eeq_Fe [21]	−0.44 V	Equilibrium potential of Fe
d_film *	1.4877 × 10^−5^ m	Electrolyte film thickness

## Data Availability

Data cannot be provided due to privacy, being this work in cooperation with a company.

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
