# Peer review of "Simulation of Corrosion Phenomena in Automotive Components: A Case Study"

_materials, 2023, doi:10.3390/ma16155368_

Round 1
Reviewer 1 Report
This work carried out a simulation and empirical study on the corrosion of galvanized steel plates. The work is interesting and worthy of publication after minor revisions.
1. Can the author supplement more abundant electrochemical experiments, such as AC impedance test?
2. Show more electrochemical experimental results, such as OCP curve, corrosion potential, corrosion current density, etc.
3. Are other tests necessary, such as SEM, XRD, etc.?
Reviewer 2 Report
This manuscript investigated the corrosion of zinc coated steel sheets by comparing simulation results with laboratory tests. Accelerated corrosion tests were conducted to examine the effects of salt fog aggression and simulate actual weathering conditions. The experimental results were then compared to the calculation results obtained from COMSOL Multiphysics. The similarities and differences between the two sets of results were evaluated to analyze the influencing parameters and refine them, aiming to find more reliable input parameters for the COMSOL Multiphysics calculations.
However, the materials and methods section lacks detailed information on the experimental procedures for the accelerated corrosion test and evaluation. It does not provide specific details such as the components or names of the equipment used, as well as the measurement methods employed to track the variation in coating thickness.
In the results and discussion section, the comparison of corrosion behavior was conducted by observing the change in thickness of the zinc coating over time. Suggestions were made to adjust simulation parameters in order to align the simulated halving time with the experimental values. However, it is important to note that the experimental and simulation results not only differ in terms of the absolute values of thickness variations over time but also exhibit distinct behaviors.
For instance, in the actual results, the thickness gradually decreases in all locations, including edges, area around the holes, and inner regions, within 80 hours. In contrast, the simulation results show rapid thickness reduction during specific intervals (specifically, between 60 and 120 hours) in the edges and area around the holes. However, in the inner regions, the thickness change is minimal and remains close to zero up to 120 hours. This raises doubts as to whether solely adjusting the simulation parameters to match the experimental results for the coating thickness halving time can adequately account for such differences in behavior.
Furthermore, in the section related to Figure 8, the authors claim that it has been possible to extrapolate the percentage of current density increment to match the experimental values of halving time. However, they have not provided the modified simulation results using this method, leaving this claim unsupported or unverified.
Therefore, in order to strengthen the author's claims, it would be beneficial to include an analysis of how well the simulation results, obtained by applying the proposed approach, align with the experimental results. Additionally, it would be helpful to provide a discussion on how the modified parameters and their values, used in this process, differ from the initial settings presented in Table 1.
Please make the following minor revisions to improve the language:
- Replace some hypercorrect or non-standard words, such as ‘apparati’ and ‘vertice’, with appropriate alternatives.
- Correct any minor errors or typos in the manuscript, such as ending a sentence without closing parentheses.
Round 2
Reviewer 2 Report
The authors have responded approrpriately to all the comments, and the uncertainties regarding the manuscript have been addressed. The revised manuscript conveys the author's claims more clearly than the previous version, and it adequately presents supporting theories or results.